# Environmental Health Diagnosis in a Park as a Sustainability Initiative in Cities

**Martha Georgina Orozco-Medina [1], Javier Omar Martínez-Abarca [1],
Arturo Figueroa-Montaño [2],\* and Valentina Davydova-Belitskaya [1]**

[1] Department of Environmental Sciences, Institute of Environment and Human Communities, University of Guadalajara, 2100 Ramón Padilla Sánchez Road, Zapopan 45200, Mexico; martha.orozco@academicos.udg.mx (M.G.O.-M.), javi_abar1296@hotmail.com (J.O.M.-A.), valentina.davydova@cucba.udg.mx (V.D.-B.)

[2] Physics Department, University of Guadalajara, 1421 Marcelino García Barragán Boulevard, Guadalajara 44430, Mexico

\* Correspondence: arturo.figueroa@cucei.udg.mx

**Abstract:** Environmental health diagnosis was made in a sport and recreational park in the city of Guadalajara, Jalisco, Mexico. The objective of this research was to perform an environmental diagnosis in Tucson Park; it concentrated on air quality (fine and coarse particulate matter, carbon dioxide), environmental noise, water quality of springs and a pond according to national standards, and macroinvertebrates as biological indicators of water condition, as well as phytosanitary state of wooded area. Additionally, a survey was conducted to study environmental perception. Results of the study highlighted significant statistical differences concerning the amount and size of particles for the winter and spring seasons. Noise levels within the park premises exceeded the Mexican standard. Water quality measured through general criteria of integrated water quality index (WQI) qualified the park´s analyzed springs as acceptable but to abstain its use and human consumption. Finally, the perception survey identified insecurity as the major problem, followed by the lack of lighting and effective communication of courses and workshops for the community to attend. It is one of the first environmental diagnoses in sport parks in Mexico´s Metropolitan Areas as an integrated approach of ecosystem health and wellbeing of city inhabitants.

**Keywords:** environmental health diagnosis; urban ecosystems; air quality; water quality index

_______________________________________________________________________________________

## 1. Introduction

From the report of the World Commission on Environment and Development in 1987, environment and development have been shown into particular actions in different domains, linked to sustainability principles, aspiring to higher parameters of quality of life without neglecting human rights, society, and its ecological processes [1]. Reflections on urban environmental quality are related to the space or territory where individuals, activities, and nature coexist [1]. Parks are prime sources of ecosystem services for city dwellers [2] and are a chief example of the urban environment. Among some of the services they provide are scenic beauty, clean air, shelter, food and nesting of birds and insects, shade, thermal comfort, areas for rest, and leisure [3].

The role of green areas in urban spaces is being increasingly appreciated [4]; the term of urban ecosystem at the end of the last century considered the primary function of green areas for societies; it included all the "green and blue spaces" that you can find in urban and peri-urban areas, including parks, cemeteries, gardens, patios, urban plots, urban forests, individual trees, wetlands, streams, rivers, lakes, and ponds [5].

In different regions of the planet and in Latin American countries in particular, green areas have gone through an intense process of urbanization that has resulted in significant impacts to the environment. The elements of the environment that have been affected the most are the quality of the air, water, and soil; in addition, it has particularly affected environmental quality in specific sites, that in turn influence social and economic development of cities [6,7].

In this sense, environmental pollution has become a serious problem for all countries, especially those in emerging economies, as reported by some studies on anthropogenic pollution in recent years [8–11]. The causes of pollution in urban areas have different ways of being studied and addressed, and some methods have included a classification relative to point sources or critical points of contamination corresponding to the areas that release higher concentrations. Second, multiple point sources, where the pollutant load comes from various point sources located in a relatively large space, make it difficult to identify the sources; as is the case of industrial parks. Finally, there are non-point sources, where the polluting load derives from the development of anthropogenic activities in large areas, for example, the use of fertilizers and the contamination of water by heavy metal industries [10,12].

The World Health Organization (WHO) estimates that each year more than 12.6 million people in the world die due to the unhealthiness of the environment, although only 847,000 of these correspond to America. It also refers, air, water, and soil pollution as a major health risk, in addition to exposure to chemicals, climate change, and ultraviolet radiation. Together, these factors are estimated to contribute to the development of more than 100 diseases or injuries [13]. In addition, the Organization for Economic Cooperation and Development (OECD) visualizes a critical panorama for the year 2050, warning that air pollution will be the main environmental problem that causes premature deaths. The estimated figure is 3.6 million deaths per year, and it also adds that emissions of Sulfur Dioxide ($SO_2$) will be 90% higher, and Nitrogen Oxides ($NOx$) 50% higher. Regarding water, it states that more than 240 million people will not have access to safe sources in terms of sufficiency, availability, and quality [14].

This discouraging scenario demands the design and application of practical public policies with an integrating approach involving society, environment, and economy, so that the environmental quality of cities is strengthened. Within these policies, an important issue is management of parks as the main source of ecosystem services for city dwellers [1–3,15]. It is out of the need for a closer knowledge of urban environments such as parks that environmental diagnosis is drawn up, which is a tool that provides methods and techniques for addressing environmental problems, thus compensating the lack of ecological and social information, that municipalities or local government entities do not provide [16–18]. Within this framework, an environmental health diagnosis is developed in a park in Guadalajara, Jalisco. The city is located in Western Mexico, this park has particular features due to its location in an area with a high influx of people, close to schools, shopping centers, and connected by primary roads. It offers different ecosystem services due to the presence of green areas, a small pond as reservoir of natural springs, sport areas such as basketball courts, volleyball, fast soccer, walking tracks, a barbecue area, and a playground for children. The results of the environmental diagnosis comprehend data on air quality, through the characterization of particulate matter, carbon dioxide, sound pressure, water quality through the development of a water quality index (WQI) based on physicochemical and bacteriological analysis, and the analysis of macroinvertebrates as a biological indicator. The phytosanitary state of trees was also performed, as well as direct interviews of users in order to know their environmental perception about the park. Completion of this project puts forward the importance of integrating socio-environmental studies in areas of conservation interest within cities, as it is the case of parks.

Environmental diagnostics in parks is a tool that allows the characterization of available natural resources, like green areas, bodies of water, as well as the evaluation of ecosystem services. Among urban ecosystems, parks provide several services, such as water and air purification, wind and noise reduction, carbon sequestration, microclimate regulation, wildlife habitat, landscape beauty, and social and psychological wellbeing [19]. Janhäll, in 2015, made an exhaustive review emphasizing numerous benefits of vegetation, confirming that tree density functions as a filter of air pollutants

[20]. Similarly, Cortelezzi et. al., in 2020, performed an environmental diagnosis under a socio-ecological approach where the pressure of urbanization on natural resources was stressed [18].

Concerning local studies, Bautista et al., in 2006, carried out a diagnosis in a neighborhood park in Mexico City. The study confirmed the importance of green areas in coexistence with urbanized surroundings to favor climatic conditions [21]. Studies in the same city of Guadalajara are those of Vázquez, in 2012, in which ecosystem and cultural services of Colomos Park were evaluated and related to social wellbeing [22]; and the book of Orozco et al., in 2018, about environmental diagnosis in cities, from which the specific air, water, and noise methods described in this paper were retrieved [16]. Lastly in the Latinamerican context, Jiménez, in 2019, carried out an environmental diagnosis in the Colombian City of Medellin, in which the description of vegetation, climatic, and environmental factors are related to climate change and health [23].

Environmental diagnosis in relation to health in public spaces is important from the ecosystem point of view, due to the influence of air and water quality, noise pollution, and the state of wooded areas that sum up to people's health and wellbeing. Air contaminants like particulate matter aggravates cardiovascular and respiratory diseases and carbon dioxide is associated with respiratory problems and causes discomfort and tiredness. The presence of noise levels causes discomfort and difficulty in communicating and concentrating, as well as physiological effects on the nervous, auditory, and cardiovascular systems. In addition, it can also affect bird communities by limiting their communication and affecting reproduction and refuge patterns. In the case of water quality, the computing of a WQI is an easy way to summarize long lists of physicochemical and bacteriological indicators of good quality standards. The presence of macroinvertebrates is a bioindicator to evaluate water quality since the presence of certain family groups only occur in certain water conditions. The health of the trees represents an opportunity for analysis due to the information it provides, to help determine the phytosanitary status that can inform on the presence of pests and foliar damage, which in turn can affect scenic beauty and photosynthetic capacity.

The aim of this investigation is to perform an environmental health diagnosis in a sport and recreational park in the city of Guadalajara, Jalisco, Mexico. It focusses on environmental variables that are well recognized in the scientific literature [5,18,24] as key elements of environmental health of ecosystems and, in turn, relevant relations to human health and wellbeing. Analyzed variables were air quality in terms of fine (0.3, 0.5, 1.0, and 2.5 μ) and coarse (5.0 and 10.0 μm) particles as well as carbon dioxide; environmental noise, water quality of springs and a pond according to national standards for water quality [25–27]; and macroinvertebrates as biological indicators of water condition, as well as phytosanitary state of wooded area. Additionally, a survey was conducted to get insights of environmental perception from park´s visitors.

In recent years, the importance of parks and urban green spaces has been getting special attention in the field of sustainable communities. Sustainability indicators for urban development to a great extent focus on the state of natural elements as it is the case of green, air, and water parameters. It is well recognized that the more native plants in an area, the greater the protection against flooding, air, and water pollution. In addition, citizens' satisfaction and perception of their living environments also play a role when developing sustainability programs that must not only be successful, but also sustainable. This paper explains the results of an environmental diagnosis concerning the state of water sources, air, and wooded area within an integrated approach of socio-environmental dimensions. These dimensions form part of the multi-dimensional structure of urban green areas evaluation that is used to improve a framework of the development and management of green areas in urban communities.

Initiatives to promote green areas in urban spaces have a positive impact on health and wellbeing of dwellers [28–30], in addition to the development of microbiomes that enhance biodiversity conservation [18,31] and lessen the effects of climate change and urban heat islands [28,32–34].

The present environmental health diagnoses match some of the Sustainable Development Goals well, as it is the case of goal 3: Health and wellbeing, goal 6: Clean water and sanitation, goal 11:

Sustainable cities and communities, goal 13: Climate action, and goal 15: Life of terrestrial ecosystems [35,36].

## 2. Materials and Methods

The park is located in a densely populated area of average socioeconomic level in Guadalajara, Jalisco, Mexico. It is one of main leisure sites with more trees in Guadalajara. It is settled into a surface of 8.3 hectares, and its floristic composition is mainly of Ash, Casuarinas, Alamillos, Eucalyptus, and others. It is located in an important area for recharging aquifers and outcropping springs, thus filling a pond and a cistern to water green areas. It has basketball and volleyball courts, as well as fast soccer. There is also a swimming pool to offer lessons, and boat hiring to take a ride along the pond. It has ample spaces for picnics and sidewalks for walking or jogging.

### 2.1. Air Quality

For air quality monitoring, 15 points were established at sites with the highest influx of visitors due to the attractiveness of its amenities. There were 15 monitoring points established to evaluate air quality in terms of particulate matter and carbon dioxide (Figure 1). Six sampling periods were carried out in total during the study. Three corresponded to the winter season (14 and 21 February, and 10 March), and the second three were completed on 2, 3 and 5 May as representative of the spring season. Morning and afternoon hours were covered during the sampling period.

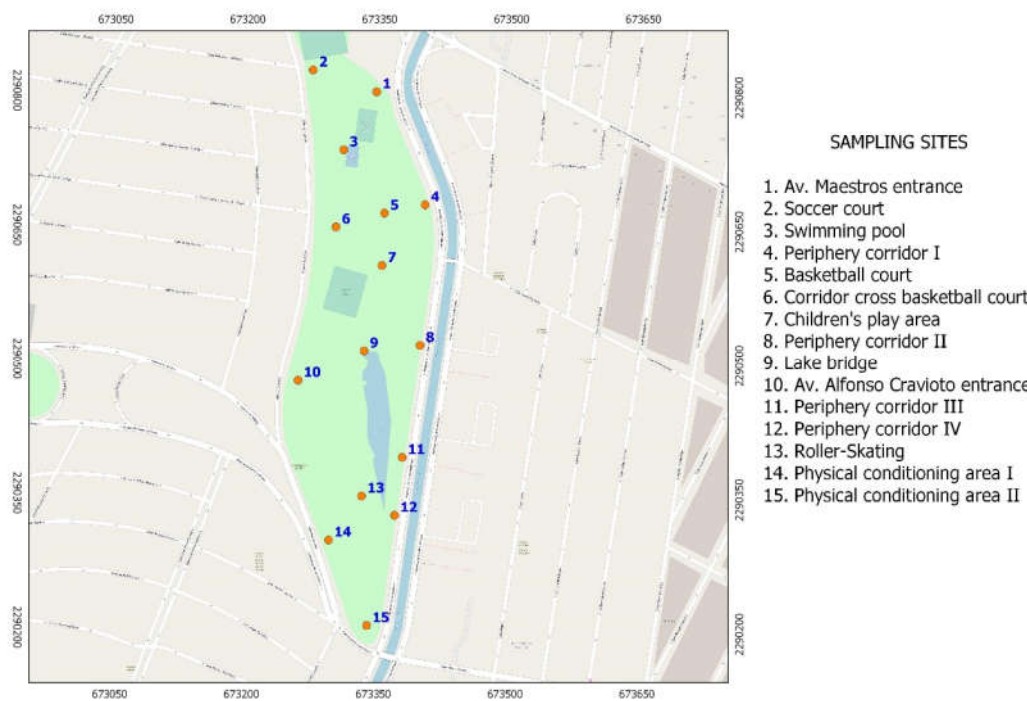

**Figure 1.** Distribution of sampling sites to monitor air quality.

### 2.1.1. Particulate Matter

For particle counting, an EXTECH® brand video particle counter was used, which performs particle counting through an isokinetic probe and differentiates the particle sizes of 0.3, 0.5, 1.0, 2.5, 5.0, and 10 μm. It has a flow rate of 2.83 L/min, and measurements were carried out in a cumulative mode by programming the equipment with five cycles at each of the sampling points. This way, the accumulation of the five one-minute cycles was added up in every point. All measurements were

performed at a respiratory height of 1.25 to 1.50 meters above the ground. Before and after the measurements, the equipment was purged using a zero-count filter, which eliminated any trace of particles that may affect new samples [16,37].

Not all the types of particles that the VPC300® measures are regulated worldwide, so the EXTECH company suggests reference values (Table 1), which have been documented in some studies [38–40]. The equipment displays particles grand total from the five cycles of 2.83 L/min. Table 2 shows risk categories according to particle size and the corresponding amount of particles

**Table 1.** Reference values for number of particles based on their size and a standard slack of 2.83 L/min.

| Size | Good | Caution | Danger |
|---|---|---|---|
| 0.3 μm | 0 a 100,000 | 100,001 a 250,000 | 250,001 a 500,000 |
| 0.5 μm | 0 a 35,200 | 35,201 a 87,500 | 87,500 a 175,000 |
| 1.0 μm | 0 a 8320 | 8,321 a 20,800 | 20,801 a 41,600 |
| 2.5 μm | 0 a 545 | 546 a 1362 | 1,363 a 2,724 |
| 5.0 μm | 0 a 193 | 194 a 483 | 484 a 966 |
| 10 μm | 0 a 68 | 69 a 170 | 171 a 340 |

### 2.1.2. Carbon Dioxide

For the $CO_2$ measurement, an EXTECH® CO250 meter was used, which has a double wavelength detector with an infrared sensor. It has a detection scale of 0 to 6000 ppm, with a resolution of 1 ppm. The maximum precision or instrumental error is ±3% of the reading. At each of the monitoring points (Figure 1), measurements of five minutes were made at a respiratory height of 1.25 to 1.50 meters. In order to ensure data reliability, the equipment should be stabilized for 10 minutes and avoid its location in front of the operator's nose [41,42].

### 2.1.3. Noise Levels

Noise levels were carried out with a CESVA DC112® integrating dosimeter with valid calibration certificate. Equivalent continuous sound pressure levels were made based on the methodology described in Orozco et al., 2015 [17]. According to the equal energy principle, the effect of a combination of noise events is related to the combined sound energy of those events. Thus, measures such as the equivalent continuous sound pressure level (LAeq,T) adds up the total energy over some time period (T) and displays a level equivalent to the average sound energy over that period. Measurements of 5 minutes were made in the LAeq mode, during the morning and afternoon when activities are carried out in the park. The equipment was placed on a tripod at 1.20 meters above ground level and at least 3.5 meters away from any sound barriers, such as walls, bridges, posts, etc., to avoid reverberation. Measurements occurred at the same points (Figure 1) and time periods stated for air quality.

### 2.2. Water Quality

The evaluation of the park´s water sources (springs) and the reservoir (pod) was assessed by the development of a water quality index (WQI) according to Bascarán [43], 1979, and Dinius, 1987 [44]. WQI is defined as the degree of contamination existing in the water at the date of a sampling, expressed as a percentage of pure water. Thus, highly polluted water will have a WQI close to or equal to 0%, and 100% represents water in excellent condition.

The main objective of computing a WQI is to turn the complex water quality data into information that is easily understandable and usable. As water quality relies on physicochemical and bacteriological parameters, these are the base of its calculation.

For the analysis of physicochemical parameters, two samplings were made according to sampling standards [45] on 1 March and 26 April as representative of the winter and spring seasons. A single sample was taken in 5-liter plastic bottles in the pond and springs (Figure 2). The bottle was immersed in the water with the neck down to a depth of 15 to 30 cm; it was immediately opened and straightened with the neck up in countercurrent. For bacteriological analysis, 250-milliliter sterile bags with a seal were used at the same time and with the procedure described above. Furthermore, extra precaution such as the use of sterile gloves, a mask, and gown by the sampler was mandatory in order to avoid germ contamination of samples. All samples were transported in a cooler to the laboratory for later analysis of coliform bacteria [27].

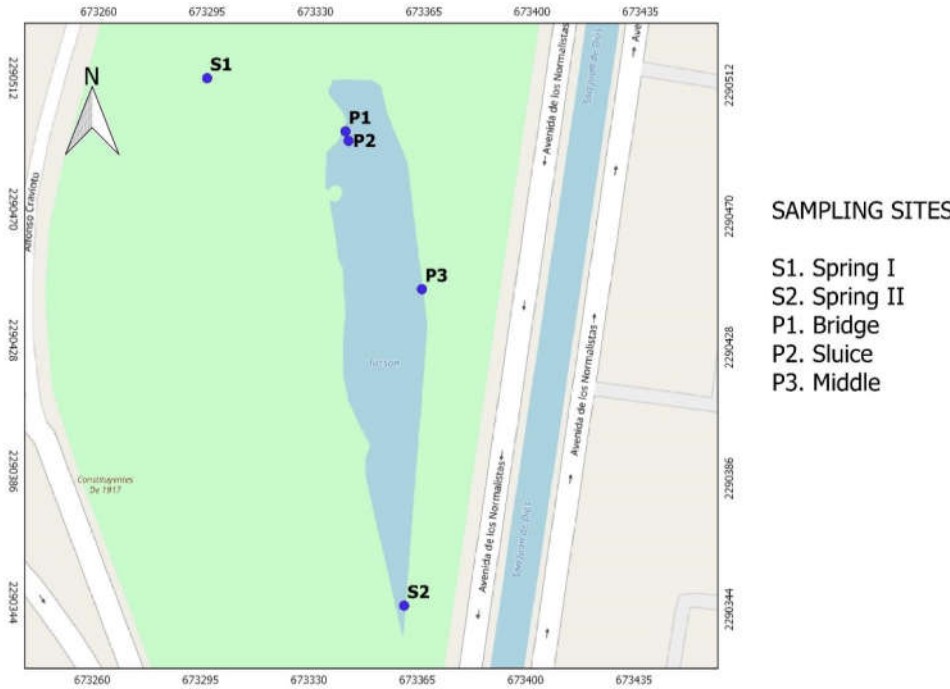

**Figure 2.** Location of water quality sites.

The pond is the reservoir of an important number of springs scattered over the park. However, water quality in this study is limited only to two representative springs (S1, S2) where visitors usually drink water. Guidelines for springs water quality were under the approach of human use and consumption [25]. Concerning the pond, the criteria was under the norm of Protection of Aquatic Life [26] since it supports aquatic animals such as ducks, fish, and turtles.

### 2.2.1. Water physicochemical parameters

The physicochemical analysis of water samples was performed according to Official Mexican Standard of water for human use and consumption [25]. Evaluated parameters are listed in Table 2.

**Table 2.** Physicochemical parameters of water and specific guidelines.

| Parameter | Unit |
| --- | --- |
| Phenolphthalein Alkalinity | mg/L $CaCO_3$ |
| Total Alkalinity | mg/L $CaCO_3$ |
| Chlorides | mg/L |
| Conductivity | μS/cm |
| Hardness | mg/L $CaCO_3$ |
| Nitrate Nitrogen | mg/L $N\text{-}NO_3$ |
| Nitrite Nitrogen | mg/L $N\text{-}NO_2$ |

| | |
|---|---|
| Dissolved Oxygen | mg/L |
| pH | pH |
| Total Dissolved Solids | mg/L |
| Sulfates | mg/L $SO_4$ |
| Temperature | °C |
| Turbidity | UTN |
| Fluorides | mg/L |
| Coliforms | NMP/100ml |

### 2.2.2. Total coliforms and fecal coliforms

The results of total coliform in the water samples were worked out by the most probable number (MPN) as it is stated in the Official Mexican Standard for the determination of coliform bacteria [27]. MPN technique is based on the ability of this microbial group to ferment lactose in a culture medium containing bile salts. Cultures are incubated at 35 °C ± 1 °C for 48 hours to reach lactose fermentation, which in turn produces acid and gas. The whole analysis takes place in two phases; first, the presence of coliforms is determined (presumptive phase), then it is determined whether the cultures that contain coliforms also contain fecal coliforms (confirmatory phase).

### 2.2.3. Water Quality Index

The products to compute the WQI are the results from the physicochemical and bacteriological analysis that will give insight into water quality (Table 3). However, as there were some aquatic animals living on the pond, a second approach under the ecological criteria of water quality for the protection of aquatic life was also included [26] and taken into account in the proposed index. Displayed colors in table 3 represent particular water conditions to communicate the degree of contamination from which decisions about its use can be taken. As a general description the blue color represents ideal conditions, the green acceptable, the yellow some restriction, and from orange to black the polluted state of water is evident. Thus is use is highly limited and health risks are significant.

**Table 3.** Classification of water quality index (WQI) according to use (Dinius, 1987) [44].

| WQI | General criteria | Public supply/Human consumption | Recreation | Fishing and aquatic life | Industry and agriculture |
|---|---|---|---|---|---|
| 100 | NO CONTAMINATED | NO PURIFICATION REQUIRED | ACCEPTABLE FOR ANY WATER SPORTS | ACCEPTABLE FOR ALL ORGANISMS | NO PURIFICATION REQUIRED |
| 95 | | | | | |
| 90 | | | | | |
| 85 | | LIGHT PURIFICATION | | | LIGHT PURIFICATION FOR SOME PROCESSES |
| 80 | | | | | |
| 75 | ACCEPTABLE | | | | |
| 70 | | | | | |
| 65 | LITTLE CONTAMINATED | GREATER NEED FOR TREATMENT | ACCEPTABLE BUT NOT RECOMMENDABLE | ACCEPTABLE, EXCEPT FOR SENSITIVE SPECIES | NO TREATMENT FOR NORMAL INDUSTRY PROCESSES |
| 60 | | | | | |
| 55 | | | | DOUBTFUL FOR SESITIVE SPECIES | |
| 50 | CONTAMINATED | | | | |
| 45 | | DOUBTFUL | DOUBTFUL FOR DIRECT CONTACT | ONLY RESISTANT ORGANISMS | TREATMENT IN THE MOST PART OF THE INDUSTRY |
| 40 | | | | | |
| 35 | | | AVOID CONTACT WITH WATER | | |
| 30 | HIGHLY CONTAMINATED | NOT ACCEPTABLE | | | |
| 25 | | | EVIDENT POLLUTION | | RESTRICTED USE |
| 20 | | | | NOT ACCEPTABLE | |
| 15 | | | NOT ACCEPTABLE | | NOT ACCEPTABLE |
| 10 | | | | | |
| 5 | | | | | |
| 0 | | | | | |

WQI occurs in three main steps. The first is to standardize individual values that are going to be used to build up the index. A variable scale from 0 to 100 is defined based on the established limit values [25,26]. A value of 100% is assumed to indicate natural or optimal conditions in the water, and 50% is the boundary between good and poor conditions. Therefore, a valuation less than 50% means that there are significant limitations for the use of water under study.

The second step is the assignment of a numerical weight to each of the parameters. The determination of the weight of a parameter (Table 4) is carried out jointly by a specialized team in water quality, and thus an estimated result is obtained for each one of them. In this work, we applied the indicative values proposed by Conesa, in 1995, for the quality of natural water [46].

**Table 4.** Weight value assigned to parameters.

| ASSIGNED WEIGHT | PARAMETER |
|---|---|
| Maximum value of 4 | Conductivity, Dissolved Oxygen, S.A.A.M. (Detergents), Color, Mercury |
| Maximum value of 3 | Turbidity, Total Phosphorus , Total Coliforms, |
| Maximum value of 2 | Dissolved Solids, Sulfates, Nitrate Nitrogen, Nitrite Nitrogen |
| Maximum value of 1 | pH, Chlorides, Temperature, Total Hardness |

Finally, the third procedure is to compute the WQI using the next equation:

$$WQI = \frac{\sum C_i \times P_i}{P_i} \times k \tag{1}$$

where $C_i$ = Percentage value assigned to the parameters; $P_i$ = Weight assigned to each parameter; $k$ = A constant value ranging from 1 to 0.25 depending on the apparent contamination of the water as follows:

- 1.00 Clear waters without apparent contamination;
- 0.75 Light colored waters, with foams and slight turbidity seemingly unnatural;
- 0.50 Waters that appear to be contaminated and have a strong odor;
- 0.25 Sewage with fermentation and odors.

## 2.2.4. Macroinvertebrates

The sampling of macroinvertebrates was carried out in a single sampling during the dry season on 8 March. The method was used for shallow water environments by Güitrón and García, 2018 [47]. Shallow water environments include rivers, lakes, and other bodies of water where it is possible to reach the bottom with your own hands and therefore with relatively small nets. It is important to use a fine mesh as many aquatic macroinvertebrates are small. Most of the studies use a mesh size of 500μm or less or type "D" nets.

In areas without flow, the net is pushed into the substrate and material is collected from the bottom. During the sampling, trawls, hitting, and removal of the substrate where the insects usually take refuge are carried out in all the micro habitats (banks with and without vegetation, submerged vegetation, leaf litter, sediment, rocks, etc.).

The macroinvertebrate classification method was the biological monitoring working party (BMWP) [48]. It looks up the family taxonomic level to determine its presence or absence and assigns a code label according to the tolerance to organic contamination of the different family groups (Table 5).

**Table 5.** Scores of the taxonomic families of aquatic macroinvertebrates for the biological monitoring working party (BMWP) index (Taken from Roldán, 2003 [48]).

| Families | Score |
|---|---|
| Anomalopsychidae, Atriplectididae, Blepharoceridae, Calamoceratidae, Ptilodactylidae, Chordodidae, Hidridae, Lampyridae, Lymnessiidae, Odontoceridae, Oligoneuriidae, Perlidae, Polythoridae, Psephenidae | 10 |
| Ampullariidae, Ephemeridae, Euthyplociidae, Philopotamidae, Athericidae, Heptageniidae, Calamoceratidae, Gomphidae | 9 |
| Gerridae, Hebridae, Helicopsychidae, Hydrobiidae, Lestidae, Palaemonidae, Pleidae, Pseudothelpusidae, Saldidae, Simuliidae, Veliidae, Odontoceridae, Lepidostomatidae, Ecnomidae, Hydrobiosidae, Leptophlebiidae, Cordulegastridae, Corduliidae, Perilestridae, Glossosomatidae, Gammaridae, Polycentropodidae, Xiphocentronidae | 8 |
| Baetidae, Caenidae, Calopterygidae, Coenagrionidae, Corixidae, Dryopidae, Glossosomatidae, Hyalellidae, Hydroptiliidae, Hydropsychidae, Leptohyphidae, Naucoridae, Notonectidae, Planariidae, Psychodidae, Scirtidae | 7 |
| Aeshnidae, Ancylidae, Corydalidae, Libellulidae, Limnichidae, Lutrochidae, Megapodagrionidae, Sialidae, Staphylinidae, Isonychidae | 6 |
| Belostomatidae, Gelastocoridae, Hydropsychidae, Mesoveliidae, Nepidae, Pyralidae, Tabanidae, Thiaridae, Polymitarcyidae, Crustacea, Turbellaria, Dytiscidae, Gyrinidae, Dixidae, Coenargrionidae, Planorbiidae | 5 |
| Chrysomelidae, Stratiomydae, Haliplidae, Empididae, Dolicopodidae, Sphaeridae, Lymnaeidae, Hydraenidae, Elmidae, Hydrometridae, Noteridae, Curculionidae, Decapoda, Simulidae, Tipulidae, Dolichopodidae, Hydrophilidae, Lygaeidae | 4 |
| Ceratopogonidae, Glossiphoniidae, Pleuroceridae, Cyclopinidae, Hidrobiidae, Cyclobdellidae, Physidae, Hidrobiidae, Erpobdellidae, Asellidae | 3 |
| Culicidae, Chironomidae, Muscidae, Sciomyzidae, Ephydridae, Thaumaleidae | 2 |
| Tubificidae, Naididae (Oligochaeta), Aelosomatidae, Syrphidae | 1 |

The sum of all families' scores compares with BMWP score ranking (Table 6) based on the principle that different aquatic invertebrates have different tolerance to pollutants. Thus, a BMWP value of 70 would correspond to slightly contaminated water of class II, with an "acceptable" quality.

**Table 6.** BMWP ranking scores and its meaning [48].

| Class | Quality | BMWP | Meaning |
|---|---|---|---|
| I | Good | >150<br>101–120 | Very clean to clean waters |
| II | Acceptable | 61–100 | Slightly polluted waters |
| III | Doubtful | 36–60 | Moderately polluted waters |
| IV | Critical | 16–35 | Heavily polluted waters |
| V | Very critical | <15 | Strongly polluted waters |

*2.3. Wooded Area*

The method used was based on a phytosanitary diagnosis adapted from Meza-Aguilar et al., 2017 [49]. Table 7 shows applied evaluation criteria from direct observation in the field. In addition, assessment of tree damage due to atmospheric pollutants was conducted as well. A group of trees at the periphery and second group of the same number inside the park were chosen at random to assess exposure to air pollutants (Figure 3). The examination was carried out from 2 to 4 May. Also, the park administration shared with us the general forest reports that they make as a requirement from the municipality.

**Table 7.** Phytosanitary diagnostics (Adapted from Meza et al., 2017 [49]).

| SPECIE: | | LOCATION: | |
|---|---|---|---|
| **Put an "x" in the corresponding characteristic** | | | |
| **TYPE** | Deciduous | | Pivoting |
| | Evergreen | ROOT TYPE | Fasciculated |
| | Branched from the base | | Superficial |
| PHYSICAL STATUS | Deformed by severe pruning | | Deep |
| | Drypoint | | Decay |
| | Dead | | Wound |
| | Exposed root | PHYTOSANITARY STATUS | Trunk tumor |
| | Dry branches | | Plague/Type |
| | Stumps | | Disease/which |
| | Cortex damage | | Pavement |
| OBSERVATION | | INTERFERENCE WITH INFRASTRUCTURE | Garrison |
| | | | Stream |
| | | | Registry |
| | | | Luminaire |

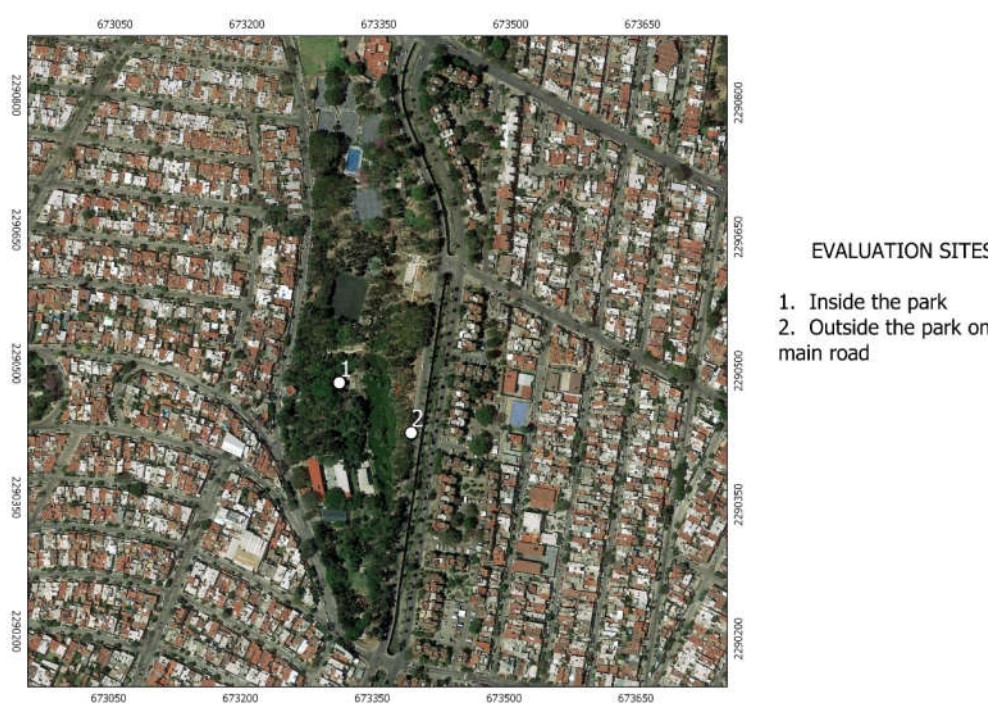

EVALUATION SITES

1. Inside the park
2. Outside the park on main road

**Figure 3.** Sites where phytosanitary diagnostics were carried out.

Inclusion criteria for tree evaluation were individuals of at least 15 cm trunk diameter at chest height, without recent pruning and without the presence of fungi on the trunk.

The analysis of the effects of atmospheric pollutants on vegetation is named "Forest Monitoring Health" and it is suggested by the United States Department of Agriculture (USDA) [50,51]. It offers

the advantage of maintaining the conditions in which the specimens grow, as it is carried out directly in the field. In this study, we looked up the evidence of harm due to ozone exposition (Table 8). It was judged on the leaves as chlorosis, necrosis, or abnormal pigmentation, and it focused on both the amplitude of damage (Table 9) and average of harm (Table 10).

**Table 8.** Damage amplitude and average severity in the plant.

| Code | Chlorosis | | | | | | Necrosis | | | | | | Reddish Pigmentation | | | | | |
|---|---|---|---|---|---|---|---|---|---|---|---|---|---|---|---|---|---|---|
| Species | 0 | 1 | 2 | 3 | 4 | 5 | 0 | 1 | 2 | 3 | 4 | 5 | 0 | 1 | 2 | 3 | 4 | 5 |

**Table 9.** Evaluation of the amplitude of damage in the plant.

| Code | Evaluation of the amplitude of damage in the plant |
|---|---|
| 0 | No harm. The plants do not present symptoms of affectation. |
| 1 | 1 a 6 % leaves witch symptoms of damage. |
| 2 | 7 a 25 % damaged leaves. |
| 3 | 26 a 50 % damaged leaves. |
| 4 | 51 a 75 % damaged leaves.. |
| 5 | More than 75 % leaves with symptoms of damage. |

**Table 10.** Assessment of damage severity.

| Code | Assessment of damage severity |
|---|---|
| 0 | No harm.The plants do not present symptoms of affectation. |
| 1 | On average of 1 to 6 % of the leaf surface presents symptoms of involvement. |
| 2 | On average of 7 to 25 % of the leaf surface presents symptoms of involvement. |
| 3 | On average of 26 to 50 % of the leaf surface presents symptoms of involvement. |
| 4 | On average of 51 to 75 % of the leaf surface presents symptoms of involvement. |
| 5 | On average more than 75 % of the leaf surface presents symptoms of involvement. |

*2.4. Environmental Perception*

An environmental perception survey was carried out to gather opinions from park´s visitors about how they value environmental conditions in the park and their consciousness of how the park´s environmental features sum up to their health and wellbeing. The survey was completed by 100 visitors that were willing to participate on the 10, 14 and 21 of March; and a second round on the 2, 3 and 5 of May. It was carried out by direct interviews following a printed format by the interviewer who gathered the information. The questionnaire comprised a total of 33 questions grouped into three sections related to general information, issues concerning visitor´s appraisal of the park environmental conditions, and questions to value the environmental state of the park in relation to health and wellbeing

The focus of the questionnaires was under a quantitative and qualitative analysis, as proposed by Hernández, in 2009 [52]. The quantitative analysis was used to determine the percentage of responses and to obtain the results of the trends in each of the questions. The qualitative approach allows a better interpretation of the environmental quality, with research strategies focused on the rigorous description of the event. These strategies are supported by observation protocols, participation, and data recordings that end up in a discussion of identified problems to build an integrated proposal to improve the park´s environmental quality.

## 3. Results

### *3.1. Particulate Matter*

With the results obtained, a multifactor analysis of variance (ANOVA) was applied to analyze if there were statistically significant differences between the means of the number of particles and the two study seasons (winter and spring), the sampling sites, and particle size. The T-student test was applied by a contrast of average effects, through the comparison of multiple ranges to identify less significant differences (LSD). This allows the formation of homogeneous groups where intra-group statistical differences are not significant. However, the differences between homogeneous values are statistically significant. The statistical software used for these analyses with a 95% statistical reliability was the STATGRAPHICS Centurion XV.

Results of the analysis showed significant effects on response variable (Number of particles) for Season and Particle size, with a *p*-value < 0.05 (Table 11). Regarding interaction, the only significant effect corresponded to Season and Particle size.

**Table 11.** Analysis of variance for particle concentration.

| Source of variation | Sum of squares | Gl | Mean square | F-ratio | p-value |
|---|---|---|---|---|---|
| MAIN EFFECTS | | | | | |
| A:Season | 2.5338E10 | 1 | 2.5338E10 | 11.04 | <span style="color:red">0.0010</span> |
| B:Site | 1.30274E10 | 14 | 9.30531E8 | 0.41 | 0.9730 |
| C:Particle size | 9.04147E11 | 5 | 1.80829E11 | 78.76 | <span style="color:red">0.0000</span> |
| INTERACTION | | | | | |
| AB | 1.33898E10 | 14 | 9.56416E8 | 0.42 | 0.9694 |
| AC | 1.41347E11 | 5 | 2.82695E10 | 12.31 | <span style="color:red">0.0000</span> |
| BC | 5.53968E10 | 70 | 7.91383E8 | 0.34 | 1.0000 |
| ABC | 5.91443E10 | 70 | 8.44918E8 | 0.37 | 1.0000 |
| ERROR | 8.26594E11 | 360 | 2.2961E9 | | |
| TOTAL | 2.03838E12 | 539 | | | |

Note: Significant *p*-value are in red font.

The multiple range test (MRT) for the Season effect differentiates two homogeneous groups where statistical difference is significant, i.e., the resulting difference among the means was greater than the LSD value (Table 12). It confirms that there is a statistical difference among means, thus highlighting the spring season with the highest count of particles in the air. Such a difference is clearly seen in Figure 4.

**Table 12.** Multiple range test for particle concentration by Season. Method: 95.0 % LSD.

| Season | Cases | MEAN | LSD | Homogeneous Groups |
|---|---|---|---|---|
| Winter | 270 | 15539.4 | 2916.17 | X |
| Spring | 270 | 29239.4 | 2916.17 | X |

| Contrast. | Sig. | Difference | +/− Limits |
|---|---|---|---|
| 1–2 | * | −13700.0 | 8110.35 |

Note: * indicates a significant difference

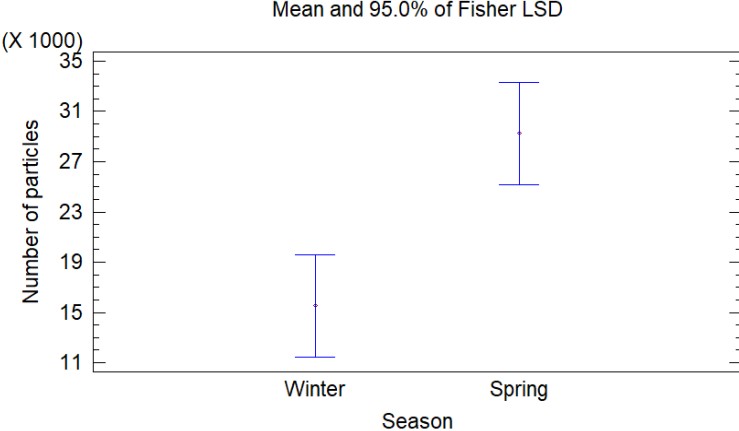

**Figure 4.** Mean differences of the number of particles per sampling season.

The second significant main effect in the ANOVA (Table 11) corresponded to particle size. Results of the MRT differentiate three homogeneous groups. The first one conformed by coarse particles (10, 5, 2.5, and 1 μm). The second and the third groups corresponded to fine particles of 0.5 and 0.3 μm, respectively (Table 13). A complementary table highlights the statistical differences among the means with an asterisk. Figure 5 shows the greatest mean of particle counts by size according to 0.3 μm. This high value can be explained from the point of view that spring is a busy season due to more favorable climatic conditions, to coexist and practice sports, and as a custom, barbecues and parties take place.

**Table 13.** Multiple range test for particle concentration by particle size.

| Particle size in μm. | Cases | Mean | LSD | Homogeneous Groups |
|---|---|---|---|---|
| 10 | 90 | 110.722 | 5050.96 | X |
| 5 | 90 | 240.022 | 5050.96 | X |
| 2.5 | 90 | 597.356 | 5050.96 | X |
| 1 | 90 | 3169.92 | 5050.96 | X |
| 0.5 | 90 | 17334.4 | 5050.96 | X |
| 0.3 | 90 | 112884. | 5050.96 | X |
| **Contrast** | **Sig.** | **Difference** | **+/− Límits** | |
| 0.3–0.5 | * | 95,549.8 | 14,047.5 | |
| 0.3–1 | * | 109,714. | 14,047.5 | |
| 0.3–2.5 | * | 112,287. | 14,047.5 | |
| 0.3–5 | * | 112,644. | 14,047.5 | |
| 0.3–10 | * | 112,773. | 14,047.5 | |
| 0.5–1 | * | 14,164.5 | 14,047.5 | |
| 0.5–2.5 | * | 16,737.0 | 14,047.5 | |
| 0.5–5 | * | 17,094.4 | 14,047.5 | |
| 0.5–10 | * | 17,223.7 | 14,047.5 | |
| 1–2.5 | | 2572.57 | 14,047.5 | |
| 1–5 | | 2929.9 | 14,047.5 | |
| 1–10 | | 3059.2 | 14,047.5 | |
| 2.5–5 | | 357.333 | 14,047.5 | |
| 2.5–10 | | 486.633 | 14,047.5 | |
| 5–10 | | 129.3 | 14,047.5 | |

Note: * indicates a significant difference.

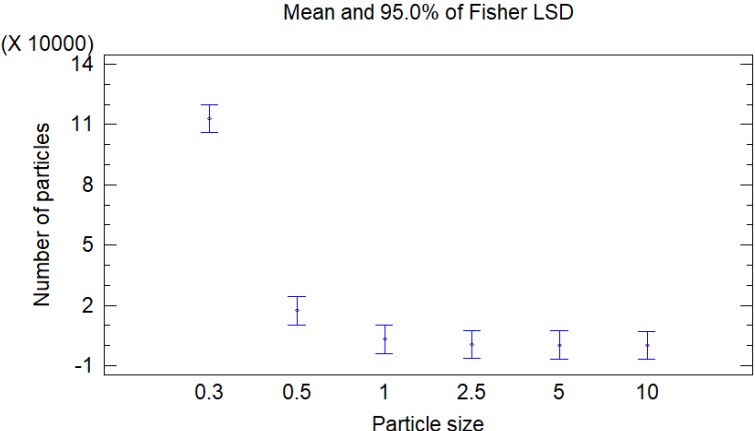

**Figure 5.** Mean differences of the number of particles with respect to their aerodynamic size.

Concerning the interaction effect by season and particle size, the most abundant number of particles were those of fine size that occurred in the Winter (Table 11, $p < 0.05$). As it is seen in Figure 6, this quantity corresponded to particles of 0.3 μm, followed by 0.5 μm and 1 μm. The number of coarse particles (2.5, 5, and 10 μm) were the lowest in the study.

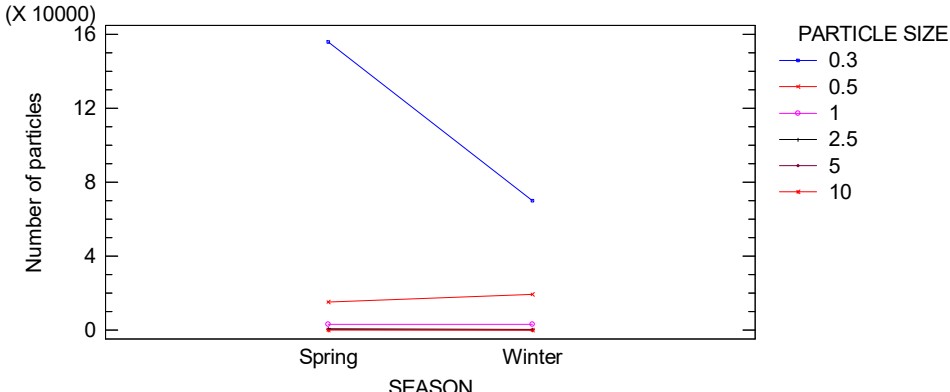

**Figure 6.** Interaction graph of number of particles by season and particle size.

*3.2. Carbon Dioxide*

In the analysis of variance, the P value indicates that there are no statistically significant differences for the concentration of Carbon Dioxide (ppm) per Season and Site (Figure 7). Carbon dioxide ($CO_2$) is produced through the natural metabolism of organisms and combustion processes. In outdoor environments, $CO_2$ levels are between 350 and 450 ppm [23].

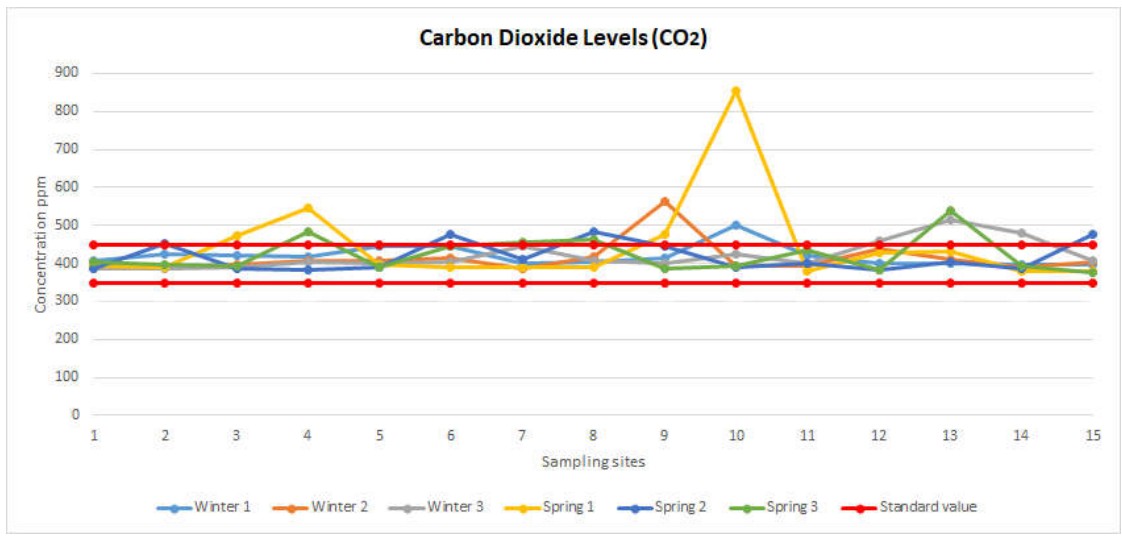

**Figure 7.** $CO_2$ concentration in different seasons.

### 3.3. Noise Levels

The analysis of variance showed the simple effect of Season to be significant for the response variable of Noise, *p*-value < 0.05 (Table 14). Regarding the interaction effect, the results do not show significant effects since the P value is larger than 0.05.

**Table 14.** Analysis of variance of Noise dB (A).

| Source | Sum of squares | Gl | Mean square | F-ratio | *p*-value |
|---|---|---|---|---|---|
| EFFECTS MAIN | | | | | |
| A:Season | 148.289 | 1 | 148.289 | 8.40 | <span style="color:red">0.0052</span> |
| B:Site | 354.95 | 14 | 25.3536 | 1.44 | 0.1652 |
| INTERACTION | | | | | |
| AB | 180.791 | 14 | 12.9137 | 0.73 | 0.7349 |
| ERROR | 1059.53 | 60 | 17.6588 | | |
| TOTAL | 1743.56 | 89 | | | |

Note: Significant *p*-value are in red font.

MRT of noise per Season denotes two different homogeneous groups where statistical differences are significant (Table 15). Mean sound pressure level was higher in winter (Figure 8).

**Table 15.** Multiple range test (MRT) of Noise dB (A) by Season. Method: 95.0 percentage LSD.

| Season | Cases | Mean | LSD | Homogeneous Groups |
|---|---|---|---|---|
| Spring | 45 | 67.3283 | 0.626433 | X |
| Winter | 45 | 69.8956 | 0.626433 | X |
| **Contrast** | **Sig.** | **Difference** | **+/− Limits** | |
| 1–2 | * | 2.56722 | 1.77209 | |

Note: * indicates a significant difference

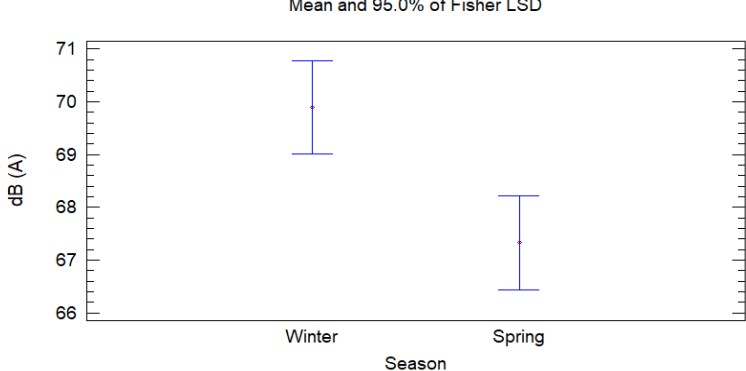

**Figure 8.** Mean Noise dB (A) by season.

In Mexico, there is an official standard NOM-081-SEMARNAT-1994 [53] in which the maximum permissible limits of noise, as well as specific hours to meet them, are established. Maximum allowed limits are 68 dB (A) and 65 dB (A) from 6:00 to 22:00 hours and 22:00 to 6:00 hours, respectively.

Noise levels in the study were higher than specified standards in Mexican legislation (black line in Figure 9) [53]. Relevant noise sources came from sport activities that usually take place with the use of high-volume music, as it was the case of aerobic and zumba dancing classes. Other important sources came from child birthday parties, basketball, and football tournaments.

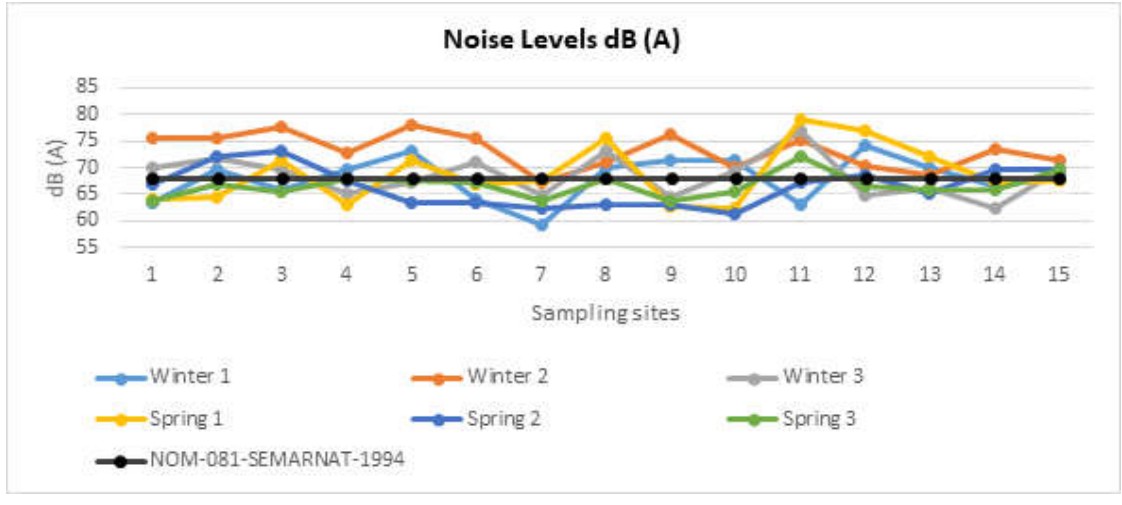

**Figure 9.** Noise levels, dB (A).

*3.4. Water Quality Index*

Higher WQI were found in the springs in contrast with the pond sites (Table 16). Although these values were above 70, according to Table 3, the use of this water for human consumption should be avoided since at this index, there is a "Greater need for treatment" and its quality according to general criteria classifies as "Acceptable". Lower WQI in the range of 26 to about 51 were registered in the pond; thus, the lowest was displayed by the spring season. Comparing the recreation column in Table 3, the pond´s water quality goes from "Avoid contact" to "Acceptable but not recommendable". This is very important regarding health risk for visitors since they practice boat riding on the pond. The lowest values at point P3 might be explained by the stagnation of water

in this site as well as the abundance of fish, the presence of ducks, and turtles, which contributed the most to the presence of coliforms.

**Table 16.** Mean values of WQI in springs and pond sites.

| Sampling | S1 | S2 | P1 | P2 | P3 |
|----------|------|------|------|------|------|
| Winter | 76.57 | 70.85 | 51.21 | 50.57 | 30.00 |
| Spring | 78.00 | 74.57 | 47.78 | 39.21 | 26.28 |

From the analyzed physicochemical parameters listed in Table 2, 33% were out of Mexican guidelines for use and human consumption [43] and 20% of results exceeded the ecological criteria for the protection of aquatic life (Table 17) [44].

**Table 17.** List of parameters above guidelines of water for human use and consumption, and protection of aquatic life.

| Water quality for Human Use and Consumption | | | | Water Quality for the Protection of Aquatic Life | | | |
|---|---|---|---|---|---|---|---|
| Parameter | Allowable Limits | Results | | Parameter | Allowable Limits | Results | |
| | | Winter | Spring | | | Winter | Spring |
| pH | 6.5–8.5 | 7.02–7.8 | 5.6–6.3 | pH | 6.8–7.2 | 7.02–7.8 | 5.6–6.3 |
| Fecal coliforms | 0 MPN/100ml | 14–230 | 23–24,000 | Fecal coliform | 200 MPN/100ml | 14–230 | 23–24,000 |
| Total coliforms | 2 MPN/100 ml | 43–1100 | 23–24,000 | Sulfates | 0.005 mg/L $SO_4$ | 58–82 | 41–78 |
| Nitrates | 10.00 mg/L N-$NO_3$ | 30–55.2 | 25.5–56.4 | | | | |
| Color | 20 Esc. Pt.-Co. | 0–167 | 0–314 | | | | |

Guadalajara has an aquifer treasure in its basement that is constantly affected by human hands. These are the springs and wells from which one out of every three liters of water that supplies citizens is removed. According to specialists, the main problem lies in the lack of public policies to conserve these aquifer deposits. All of this has not been addressed in a sustainable way and the result is that the levels of most of the city´s water tables are decreasing or have been contaminated by leaks due to intense economic activities as the capital city of the State of Jalisco [54].

Macroinvertebrates

The identification of macroinvertebrates during direct collection in the pond showed the following taxonomic families and scores for the BMWP method (Table 18):

**Table 18.** Taxonomic families and related score.

| Family | Score |
|--------|-------|
| Cyclopinidae | 3 |

| | |
|---|---|
| Chironomidae | 2 |
| Leptophlebiidae | 8 |
| Elmidae | 4 |
| Hidrobiidae | 3 |
| Chrysopidae | 5 |
| Lymnaeidae | 4 |
| Hyalellidae | 7 |
| Oligochaeta | 1 |
| Tabanidae | 5 |
| Thiaridae | 5 |
| Pleuroceridae | 3 |
| Decapoda | 4 |
| Simuliidae | 8 |
| **BMWP** | **70** |

According to BMWP ranking scores (Table 6), the resulted BMWP matched a class II, an acceptable water quality, which corresponds to slightly polluted waters.

*3.5. Wooded Areas*

There are 1848 trees in total, 98% of which are in good condition, but most have some kind of pest (Table 19). A fundamental part of doing phytosanitary evaluation is to include the organisms that are in the periphery, since they also influence the health of internal organism.

**Table 19.** Phytosanitary state of wooded area.

| Total Trees | Good Condition | Dry | Pest Infestation | Hollow | Healthy at Risk of Falling | Stumps |
|---|---|---|---|---|---|---|
| 1848 | 1823 | 15 | 90% | 7 | 2 | 101 |

Based on the analysis carried out and the classification that the USDA suggests [38], at least 6 different affectations of the park's trees can be listed, both on trunks and leaves.
Trunk:

- Canker: Dead, discolored, and often sunken area of a plant.
- Tumors: Caused by the parasitic proteobacterium *Agrobacterium tumefaciens.*
- Mistletoe: *Viscum album* is a semi-parasitic plant that penetrates the bark of trees. It absorbs water from the branches and trunk, mineral salts, and nutrients that it cannot obtain by itself. In cases of severe infestation, it can kill the tree.

Leaves:

- Powdery mildew: It is a cryptogamic disease produced by several genera of ectoparasitic fungi that mainly attack young leaves and stems. Its main sign is the appearance of a mealy or cottony layer and a white or grayish color, formed by the mycelium and the conidia.
- Necrosis: Death of tissue, usually accompanied by black or brown darkening.
- Chlorosis: it is an abnormal physiological condition in which the foliage produces insufficient chlorophyll. When this occurs, the leaves do not have the normal green coloration; the coloration is pale green, yellow, whitish yellow.

All these diseases and pests were evident in many of the park trees, both within the premises and on the outer periphery, thus assuming that, due to their proximity and lack of space, any diseased tree is a source of contagion to surrounded healthy wooden area.

### 3.6. Environmental perception

The survey was applied to 100 volunteers who accepted to participate during the environmental sampling days. The survey results highlighted that the majority of people who attended the park were local visitors (61%), followed by athletes from nearby neighborhoods (32%). The age distribution was 33% for those between 15 and 20 years old, and 27% for those between 21 and 30 years old. Regarding sex, half of the respondents were men and the other half women, 68% of them mentioned living in the Municipality of Guadalajara, and 20% in Zapopan. Regarding the residence time, the majority mentioned more than 20 years (33%), followed by 10 to 15 years (30%), and in relation to their occupation, identified themselves as students and workers at 44% and 41%, respectively. Among the students, 39% reported undergraduate studies and 30% high school. The frequency of visits was 2 to 3 times a week to occasionally (23%), weekly (22%), and daily (17%). Regarding infrastructure, 24% demand greater security, 23% garbage separation, and 13% clean toilets. Thirty-two percent of the respondents indicated that they assist in recreation and to have fun and 28% to relax. Regarding the problems that most drew their attention, they are neglect of green areas and insecurity (17%) and among the benefits they perceive when visiting the par was health with 21%, 17% tranquility, 15% fun, and 60% agree that the park administration corresponds to the municipality.

The questionnaire comprised a total of 33 questions grouped into three sections related to general questions, issues concerning visitor´s appraisal of the park environmental conditions, and questions of valuing the human–environment relationship based on public health as suggested by Seymour (2016) [55].

For the purpose of this project, only questions related to the relation between visitors and the environment are emphasized in pie charts. Figure 10 shows the results of visitor´s appraisal of the park´s environmental conditions.

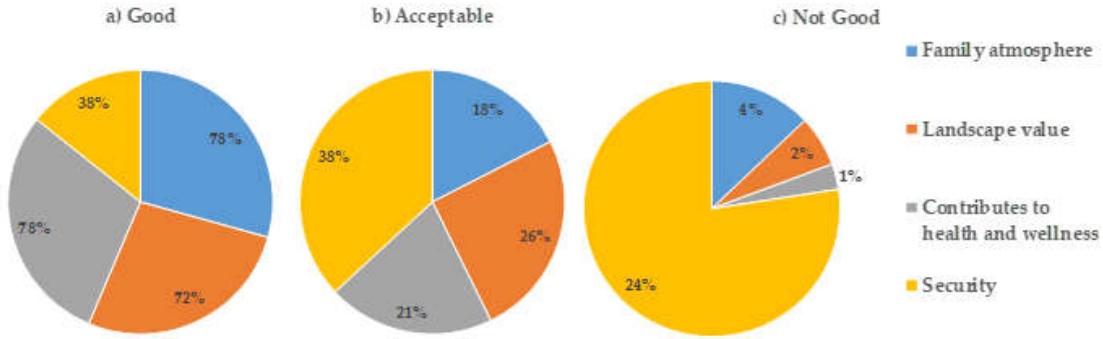

**Figure 10.** Environmental and health perception of park visitors.

It is clear that the majority of survey participants identified three main positive contributions as good from park environmental features. These were park contribution to their health and wellness, family atmosphere, and landscape beauty. However, such good features are overwhelmed by security issues within the park premises. The lack of security in urban parks presents serious problems and devalues the great environmental services one can benefit from by visiting parks and similar open leisure facilities.

Interviewed visitors identified that the most benefit they get from visiting the park was related to welfare, as it is the case of health, calmness, fun, and happiness (Figure 11).

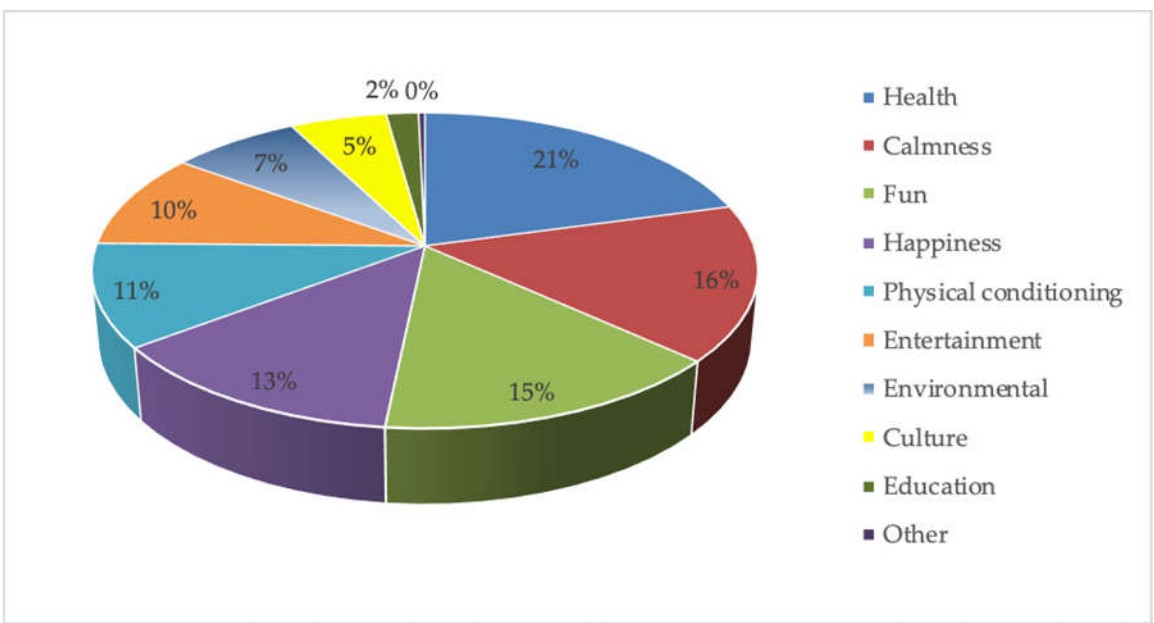

**Figure 11.** Benefits perceived by the park visitors.

## 4. Discussion

Results from the environmental health diagnosis as a sustainable initiative in cities highlighted the presence of important health risks. In terms of sustainability, parks are a model to apply the different strategies that allow actions to be taken into conservation of natural resources. Characterization of park resources helps to know the state of occurring natural elements from which integrated policies can be put into practice to preserve value and ecosystem services. In the analyzed park, components of great value are water resources like scattered natural springs occurring within the park premises, and the pond, which is the reservoir of them. The wooded area is also important since trees function as natural air filters, shelter, bird nesting, soil stabilization, and microclimate regulators as they lessen the effects of urban heat island.

Concerning air quality, fine particles in the range of 0.3–0.5 μm were higher during the spring sampling than in the winter. Particle matter varies significantly on time and space, due to the dynamics of meteorological conditions, the heterogeneous properties of the earth's surface, the uneven distribution of emission sources, geographic features, and human activities [20,55]. In a global scale, it is the most prominent pollutant affecting air quality in urban areas especially in developing countries [56].

Depending on their size, the particles are categorized as coarse, which include particles with a diameter between 2.5 to 10 μm. These particles that generally rise from the ground hardly penetrate into the pulmonary alveoli, since for the most part, they are retained by the mucosa and cilia of the upper part of the respiratory system. In contrast, particles from forest fires and agricultural burns, as well as those generated by motor vehicles, are mostly fine (<2.5 μm) and have the capacity to penetrate the lung alveoli. In addition, particles can have toxic effects due to their inherent physical or chemical characteristics, or they can indirectly affect humans both by interfering with mechanisms of the respiratory system and by acting as a vehicle for transportation of toxic substance coated on its surface, that might be distributed into the body systems [57].

Particles represent a serious health problem in cities and have been associated with increased mortality due to heart and respiratory diseases as well as cancer and strokes. Mortality rates depend on type of exposure and the length of time, as well as particle size. Fine particles have the power to trespass alveoli membranes to enter the bloodstream and then get distributed systematically to the heart, kidneys, bladder, and another important organ where they can exert toxic effects [58–62].

The area in which the park is located is one of the busiest in Guadalajara, Av. de los Normalistas. It is possible that the greatest number of fine particles registered was due to gas

emission from the combustion of motor vehicles. The greater number of fine particles in the spring sampling season can be explained in terms of favorable weather condition that invite families to spend leisure time in the park. Furthermore, it is a cultural custom in Mexican families to have barbecues in picnic areas of the park. Barbecue fumes are fine particles [63,64]. In addition, regional atmospheric circulation carries the smoke to the city, resulting from agricultural burning of land preparation before the rain season. It is well documented that fumes from this practice can travel hundreds of kilometers towards urban areas when the winds are favorable. It is important to comment that agricultural burning is a traditional activity of Mexican farmers in early spring as it is the cheapest procedure to get rid of products that remain in the field after harvest. [65–67]. It is also true that in spring, temperature rises significantly, as Romero and Sarricolea, in 2006, found a linear relationship between atmospheric temperatures and particulate matter, in addition to a strong correlation among soil use and cover on temperature elevation, which can exacerbate health effects of chronic diseases like cardiovascular and respiratory illnesses [68,69]. It is worth mentioning that the track where athletes run is not paved completely and it is a risk since both fine and coarse particles can be suspended in the air by wind currents that then enter the athlete's respiratory system.

Carbon dioxide ($CO_2$) is produced through the natural metabolism of organisms and combustion processes. For the analysis of $CO_2$, several authors have suggested different reference limits for its measurement; in indoor environments, $CO_2$ levels vary between 400 and 2000 ppm, while $CO_2$ levels outdoors are 350 at 450 ppm [40]. The health hazard information sheet proposed by the United States Department of Agriculture (USDA) and Environmental Safety and Health Group (ESHG) indicates that $CO_2$ levels in outdoor air range from 300 and 400 ppm as normal limits, but can be as high as 600 to 900 ppm in metropolitan areas [70]. $CO_2$ emissions from the energy and transport sector together with the waste management sector are responsible for the generation of greenhouse gases. Industrialized countries contribute approximately 80% of the total load, therefore special attention should be given by governments [71].

Regarding $CO_2$, one possibility that we can delve into is the estimation of carbon dioxide stored in trees that has important variations depending on the species, growth patterns, and age of tree [72]. When people are exposed to $CO_2$ levels between 800 to 1200 ppm, they experience discomfort and symptoms related to headaches, tiredness, breathing problems, drowsiness, difficulty concentrating, dizziness, and eye irritation. Such health signs are usually aggravated in children and the elderly [73]. The Institute of Security and Hygiene of Work from Spain recommends that $CO_2$ levels should be reduced to less than 600ppm to eliminate discomfort [74].

$CO_2$ concentrations in this study did not show statistical differences due to season and sites. The peak registered at site 10 in Figure 7 exceeded Halgamuge permissible limits [42] and can be explained by a barbecue event at the time of sampling, since grill infrastructures are nearby this sampling site, and coal combustion is $CO_2$ reach. However, it is difficult to say that the registered level was solely due to the mentioned event since in terms of air pollution, local and regional patterns of atmospheric circulation also play a role.

Regarding noise levels, it was found that they were much higher than WHO [75] and EPA [76] recommendations of 70 dB (A) for traffic areas. Noise pollution has a significant contribution to the health and wellbeing of city dwellers. It has been related to cardiovascular, gastrointestinal and nervous diseases, tachycardia, decreased appetite, insomnia, irritability, stress, affects communication, and represents difficulties in focusing and the performance of academic tasks, in addition to the most common effect, which is hearing impairment [77,78].

Martínez and Moreno in 2013 stated that noise studies in parks are scarce in the scientific literature. In the study, they present results from five parks in the City of Madrid in Spain where they found average levels from 63.9 to 68.9 dB (A) and minimum levels around 54 to 62 dB (A) [79]. Alfie and Salinas, in 2017, reported the benefits of parks as they observed a noise decrease of about 5 dB (A) in pedestrian walkways in Mexico City [80]. A study performed in Santiago de Chile by Pltazer et. al 2007 described levels of around 58.6 dB (A), which were significantly lower than 110 dB (A) in bars and the underground public transport where they recorded 87.1 dB (A) [81]. Another

study in public spaces in Hong Kong, China indicated average levels above 65dB (A) [82]. Brambila and Maffei, in 2006, studied noise in Naples parks where they evaluated responses to noise, and 57% mentioned valuing sounds as the most important features in the parks [83].

In Mexico, there is an official standard NOM-081-SEMARNAT-1994 [54] in which the maximum permissible limits of noise, as well as specific hours to meet them, are established. Maximum allowed limits are 68 dB (A) and 65 dB (A) from 6:00 to 22:00 hours and 22:00 to 6:00 hours, respectively. The presence of natural sounds in parks is a treasure that must be value. Kogan et al., in 2014, describe the benefits of renewal of cognitive and emotional health that eventually promote healthy psychological and physiological processes, hence the importance of keeping urban parks in good condition [84].

Maximum levels of up to 79 dB (A) were recorded at point 11 during the spring season, therefore the area is considered to be slightly critical, but still has the potential to cause damage to exposed individuals. The main effects relate to the failure of concentration and communication, alteration of the auditory system, among many other health problems related to noise exposition [16]. However, the statistical analysis by site indicated no significant differences. The analysis of variance by season highlighted significant differences between winter and spring, with the former having a higher mean effect on noise levels. Such a difference might be due to the fact that during the samplings there were more events with music, child birthday parties, and the pruning of gardens.

Water pollutants were of interest in this study because of their importance to health risks due to the presence of microorganisms that can develop contaminated water vector diseases such as gastrointestinal illnesses, and they were evaluated according to national standards. Sources of water contamination are diverse and might come from domestic and industrial waste waters that alter physicochemical properties and bacteria load or add toxic substances such as acids and heavy metals that, in high concentrations, will limit development of aquatic life [85,86].

The state of water sources in the park were evaluated through a computed WQI that provided an easily understandable scale to qualify water quality. The products to develop the index are physicochemical and bacteriological parameters that must be grouped together to come up with a representative scale indicative of the level of contamination [87].

Points located in the pond (P1, P2, and P3) recorded lower WQI due to stagnant conditions of water at this point. Moreover, there is a great abundance of fish and ducks in the pond and might be the explanation of the high amounts of coliforms. Adding factors to these values might be pollution sources due to residues of aquatic fauna, bad behavior of some park visitors throwing garbage into the pond, and improper feeding of aquatic fauna.

Spring points S1 and S2 showed the higher WQI. Although its water is not suitable for use and human consumption, as WQI categories stated a "Greater need for treatment", ignoring these results will result in a significant increase in gastrointestinal diseases. This result is relevant since visitors used to drink water from this site. Untreated domestic waters and storm runoff from agricultural and urban areas usually contain high concentrations of nitrogen and phosphorus, and these nutrients are essential for the growth and development of phytoplanktonic organisms such as green algae, cyanobacteria, or diatomic algae. However, excessive input of nutrients into the water can trigger the growth of phytoplankton. When phytoplankton dies, it is broken down by heterotrophic bacteria that, in the process, consume oxygen. When the amount of dead phytoplankton is very abundant, the bacterial activity increases remarkably, reducing the oxygen level in the water. This produces, among other effects, bad odors and the death of aquatic species that need well-oxygenated waters to survive, thus unbalancing the natural drainage of water [88].

The supportive criteria to evaluate water quality by the presence of certain macroinvertebrate families in the water was through the BMWP index, which is widely used in various countries due to its validity. It can generate information regarding the characteristics of ecosystems with their respective hydrological and ecological adaptations to particular environments. The reliability of index makes it an excellent tool for monitoring and managing watersheds, as well as for research studies, which are especially useful when there are no regulations in this regard [48,89].

Macroinvertebrate findings labeled the pond´s water as "slightly polluted" due to the presence of Pleuroceridae, Chironomidae, and Cyclopinidae families that can be found in not very pure waters. Matching of this result to WQI was close since the corresponding category was "Acceptable but not recommendable" for performing recreational activities on the pond, as it is the case of boat riding. It is worth mentioning that the use of macroinvertebrates as bioindicators of water quality in the Mexican and even the Latin-American contexts are scarce, therefore results of the present study will make a relevant contribution in this region and will add knowledge to the global context [90].

The health of wooded areas is considered to be of great importance in parks since healthy trees are strong trees that promote photosynthesis, contribute in a better way to the uptake of pollutants and the generation of oxygen, they are not at risk of falling, and therefore its ecosystem and health role is most effective in parks. The benefits of green areas for the population are duly documented in terms of physical and mental health [91–93]

Evaluation of wooded areas confirmed similar results concerning chronic damage that has been reported by Gallegos (2013) where it was pointed out that not all observed were due to ozone exposure, but rather ensuring a relationship between different atmospheric pollutants with the presence of pathogens or pests, which seriously affects the health of the tree. In our study, the majority of trees (90%) suffered from diseases and pests that put their presence in the park at risk. The problem is widely distributed among wooded area, i.e., within the premises and on the periphery of the site, assuming that, due to their proximity and lack of space, any diseased tree represents a potential source of infection and easy spreading of disease [94].

Environmental perception surveys are a very useful instrument since they can be used to identify the inconvenience and complaints of the population regarding exposure to adverse environmental conditions, insecurity, or problems in general that limit their wellbeing, and also support the implementation of programs, projects, and actions that can be undertaken to improve the operation of public places, such as parks in this case [95,96].

Answers from the environmental perception survey emphasized that park users were not totally aware of the main pollution problems in the park, since they were only able to identify the pond´s contamination by the bad smell and particular color of water. This puts forward the intensive work that must be done in society to awake people´s consciousness about the relationship between the state of the environment and self-welfare as it is one of the principles of sustainability. In a general context, people evaluate pollution problems based on visible features and fail to visualize the problem in an integrated socio-environmental approach as they lack the skills to do it. In this context, survey participants mentioned that the park is in good condition, thus referring to infrastructure and amenities. Their suggestions were focused on infrastructure remodeling of playground areas and the running track due to its evident weathering. Special request was also put in for solving the alarming situation of insecurity and lack of public lighting. In addition, they requested better dissemination of the activities carried out in the park, since they do not know the services offered by the Municipal Sports Council.

## 5. Conclusions

Different investigations guarantee that environmental diagnosis is the best method to know the quality of the environment in a complete and complex way, which helps to identify the main problems of ecosystem deterioration [18,97,98]. It encourages participation for designing environmental research models that will end up in comprehensive public policies, thus incorporating different disciplines in a multidisciplinary approach of environmental health. In the same way, it retrieves the value of ecosystem as a multifunctional space, where parks offer basic services that must be considered when planning their development and conservation.

Therefore, the results of this work are considered important, both in terms of air and water quality of the park, and the perception of the importance of public spaces such as parks or urban forests for public health. Among these, we can highlight:

- CO$_2$ was detected within the limits of permissible levels for human health, which can be justified by the abundant presence of trees and vegetation in general, which purify the air by absorbing this pollutant as input to the processes of photosynthesis.
- The high levels of particles of 0.3 μm in spring that can be explained in terms of favorable weather conditions that invite local residents to spend time visiting the park and the opportunity for family and friends gathering, which is usually accompanied by barbecues. Barbecue fumes area reach fine particles. In addition, regional atmospheric circulation carries the smoke to the city, resulting from agricultural burning of land preparation before the rain season. It is well documented that fumes from this practice can travel hundreds of kilometers towards urban areas when the winds are favorable. Park location might also play a role, since it is nearby main avenues and busy roads.
- The inadequate use of equipment that produces music in public spaces to accompany sports activities, family, or school gatherings is usually the cause of high noise levels that exceed national regulations [53].
- As a negative result, and of concern from the point of view of public health, was the analysis of water quality, and detection of different types of coliforms (total and fecal) was above the respective standard limits.
- BMWP index of macroinvertebrate analysis labeled the ponds water quality as "slightly polluted" due to the presence of Pleuroceridae, Chironomidae, and Cyclopinidae families that can be found in not very pure waters. Matching of these results to WQI was close since the corresponding category was "Acceptable but not recommendable" for perform recreational activities on the pond.
- From the survey about possible benefits of visiting the park, the categories of health, calmness, fun, and happiness accumulated 65% of respondents' options. In general, the importance of visiting the park was focused on physical and mental health.
- Finally, the infrastructure of the park is in good condition, proper maintenance and monitoring would be a good option to prevent deterioration. In most answers of the survey, the lack of security was indicated, as well as lack of lighting and dissemination of courses and workshops for community participation. Survey results from the study highlighted perception surveys as a proper tool to identify environmental problems of great value from which a comprehensive intervention proposal can be designed to tackle identified problems. It is worth mentioning that users have a good impression of the site, because places like this for leisure are scarce in Guadalajara.

Finally, it is important to recognize that practically all the trees in the park are sick or have some type of damage. All analyzed park trees have had or have a disease or pest. Driving forces might be excessive pruning, such as cutting branches, cutting the trunk completely, or leaving it to die. This type of problem was not noticed by visitors and probably not by the park´s administration. However, it is important to emphasize that the health of trees in cities is synonymous for good or bad environmental health dependent of the state of wooded areas.

**Author Contributions:** Conceptualization, J.O.M.-A.; data curation, V.D.-B.; formal analysis, A.F.-M.; funding acquisition, M.G.O.-M.; investigation, J.O.M.-A.; methodology, M.G.O.-M.; project administration, M.G.O.-M.; resources, V.D.-B.; supervision, M.G.O.-M.; validation, V.D.-B.; writing—original draft, J.O.M.-A.; writing—review and editing, A.F.M. All authors have read and agreed to the published version of the manuscript.

**Funding:** This research was funded by Nacional Council of Science and Technology (CONACyT, Mexico), grant number 25755. The equipments used in this research were acquired by the Institute of Environment and Human Communities (IMACH) through Institutional funding from the University of Guadalajara.

**Acknowledgments:** The authors of this article would like to thank the colleagues of the Department of Environmental Sciences, Campus of Biological Sciences and Agriculture, University of Guadalajara. Special thanks go to Javier Garcia Velasco for his support in field work and advice for physicochemical analysis of water samples; Josefina Casas Solis for her support in microbiological analysis of water samples; María Marcela Güitrón López for suggesting the methodology to study macroinvertebrades as biological indicators of water

quality. Finally, we would like to thank the students: Anayeli Chavarin, Ángela Santiago, Alejandro Mendoza, Alondra Benitez, Nataly Castillo, Cesia Torres, Kenia González, Denisse Sánchez, Héctor Preciado, Gabriel Torres, Andrea Godínez, Edgar Mejía and Jesús Alvarez for their suppot in field work and the interviews to Tucson Park users to complete the study. Julieta Delgadillo Orozco for the translation of original manuscript into English. We also thank the Park´s management office for the permission and information given concerning the care of wooded area.

**Conflicts of Interest:** The authors declare that there is no conflict of interest regarding the publication of this article. The founding sponsors had no role in the design of the study; in the collection, analyses, or interpretation of data; in the writing of the manuscript, and in the decision to publish the results.

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
