# Peer review of "Environmental Health Diagnosis in a Park as a Sustainability Initiative in Cities"

_sustainability, doi:10.3390/su12166436_

Round 1

Reviewer 1 Report

The manuscript is very interesting and valuable topic for providing healthy environment to urban dwellers.

  1. Need more explanation on environmental diagnogis, in term of definition, any other cases implemented, etc.
  2. Need more explanation on the variables in  the environmental diagnogis, why these are important to humahealth and why park contains the optimal levels of the variables.
  3. Provide a table or checklist to monitor the environmental diagnogis.

Author Response

We appreciate all your comments which we respond to each of them in the attached file. In the same way, each one of them is included in the revised version of the manuscript.

Reviewer 2 Report

The paper presents results obtained from systematic research into the environmental issues in one of the municipal parks in Guadalajara. It presents interesting insight into the current problem of the environmental health diagnosis of public green space. It is also commendable that the Authors used statistical tools to analyze the results. However there are several issues.

Major remarks:

  1. The paper doesn’t have a well-defined aim. What can be considered to be an aim (lines 91-92) is very vague and does not show any particular research interest of the paper. The Authors should define the aim of the paper in a clear, concise way.
  2. While the results are interesting, the discussion is rather rudimentary, and does not touch upon what exactly is there to gain from obtained and presented data. “Sustainability” is a journal of a worldwide reach, and therefore it would be beneficial to present why the results should be of interest to the worldwide audience. In its present form, the paper seems more like a report from the research, and less like an article.
  3. Lack of aim has another drawback – the conclusions vary from too generic (and not resulting from presented research) like lines 511-512 or 539-545 (which could be the introduction of the conclusions, if anything), to way too specific (e.g. line 527), with little to none synthesis of the obtained results. The Authors should reconsider the conclusions, in connection to the aim of the research.
  4. In the materials and methods section, it should be made clear in each case how many measurements were taken. It is well described in the 2.1. Air quality (lines 104-107), however there is no such straightforward information about the water quality tests or tree evaluation.
  5. In the discussion (lines 418-422), Authors attribute the springtime increase in particles in air to pollination – however, it is stated in the paper that pollination ranges from 2-100 μm, while the fraction in question is 0.3 and 0.5 μm. This appears contradictory, and might be worth commenting upon.  

Minor remarks:

  1. Authors present survey data (381-399), however it is visibly incomplete. It may be better to show full results (or the most important questions) of the survey either as a table, or charts. When presenting a survey, it is also beneficial to present the exact questions and set answers to close-ended questions, as well as a way the survey was conducted (being given a paper to fill out, being asked the questions by the research staff, sending the answers through e-mail, etc). It allows to better gauge what the results indicate.
  2. I would suggest that it may be beneficial for this study to include comparison data when applicable, for example by finding similar research into parks from different cities.
  3. The is no need to present photos of the equipment that was used in the research.
  4. Are Authors sure that the use of ANOVA for data which is in logarithmic scale is correct?
  5. The abbreviation TP for the name of the park is used only once in the whole text. Is it necessary to use this abbreviation?
  6. Some references are described wrongly, e.g. line 560, there should be  author and title of linked paper, similarly 580, 636, 673 where is just an organization name and no title. Also, the “Disponible en” should be changed to English.

Author Response

(The authors gave the same response as above.)

Round 2

Reviewer 1 Report

The authors revised the manuscript and it has improved.

Author Response

We thank your comments about the improvements made in the revised version of the manuscript. However, as we did not find any specific comments about question 2, 3 and 4 of the Review Report Form, which you marked as "can be improved", we were not able to provide any extra information from the one we reported in the first round.  

Reviewer 2 Report

I would like to thank the Authors for providing the in-depth answers to the issues I had with the paper, and thoroughly addressing them in the paper. Vast majority of the problems I’ve noticed was fixed, however still some issues persist:

MAJOR REMARKS:

  1. While the aim of the paper was rephrased and amended so that it is clearer, there is still no mention of why this issue should interest the reader who is not from Guadalajara (or a visitor of the park in question). “Sustainability” is a journal of worldwide reach, and while the research provides an insight important for the park’s use and maintenance, the paper provides no reason why this issue is important to a wider audience. Please add to the introduction information why this research is significant for a wide audience. It it’s present form, the paper still resembles a technical report instead of an article.

MINOR REMARKS:

  1. While I thank the Authors for in-depth answer to my question of logarithmic scale, it was not about the particle size, which as Authors said, it is absolutely correct to use ANOVA for, but it was about the sound levels. Difference of 3dB which was between the seasons amounts to almost doubling the sound energy, so there is no issue whether or not there is a difference between noise in seasons, as it is significant. However please consider the formal use of ANOVA for dB measurements, as even 1 dB of difference is a significant change of sound intensity heard on site. 
  2. There’s a formatting issue with a part of tab. 3 in line 248

Author Response

We appreciate all your comments which we respond to each of them in the attached file. In the same way, major remarks are included in the revised version of the manuscript.
